# Converse flexoelectricity yields large piezoresponse force microscopy signals in non-piezoelectric materials

Amir Abdollahi [1], Neus Domingo [2], Irene Arias [1] & Gustau Catalan[2,3]

Converse flexoelectricity is a mechanical stress induced by an electric polarization gradient. It can appear in any material, irrespective of symmetry, whenever there is an inhomogeneous electric field distribution. This situation invariably happens in piezoresponse force microscopy (PFM), which is a technique whereby a voltage is delivered to the tip of an atomic force microscope in order to stimulate and probe piezoelectricity at the nanoscale. While PFM is the premier technique for studying ferroelectricity and piezoelectricity at the nanoscale, here we show, theoretically and experimentally, that large effective piezoelectric coefficients can be measured in non-piezoelectric dielectrics due to converse flexoelectricity.

[1] Laboratori de Càlcul Numèric (LaCàN), Universitat Politècnica de Catalunya (UPC), Campus Nord UPC-C2, E-08034 Barcelona, Spain. [2] Catalan Institute of Nanoscience and Nanotechnology (ICN2), CSIC and The Barcelona Institute of Science and Technology, Campus UAB, Bellaterra, 08193 Barcelona, Spain. [3] ICREA-Institut Catala de Recerca i Estudis Avançats, 08010 Barcelona, Catalonia, Spain. Correspondence and requests for materials should be addressed to A.A. (email: amir.abdollahi@upc.edu) or to N.D. (email: neus.domingo@icn2.cat)

Piezoresponse force microscopy (PFM) has become the go-to technique for characterizing piezoelectricity at the nanoscale (particularly in ferroelectric thin films) and manipulating ferroelectric polarization[1,2]. It is also often used as a tool to aid establishing whether or not materials are ferroelectric or piezoelectric; however, as has been pointed out[3–8], this latter use is delicate, because there are other physical phenomena that can yield a piezoelectric-like response in PFM without the material in question having to be piezoelectric.

PFM operates by delivering a voltage $V$ to the surface of the material via an electrically conducting tip in an atomic force microscope (AFM). However, the application of the conductive tip directly on the sample leads to an effective piezoelectric coefficient that may not coincide with the intrinsic piezoelectric coefficient of the material. Even in the ideal case of a homogeneous and insulating piezoelectric, the electric field itself is not evenly distributed across the material, so the measured piezoelectric coefficient is an average of the field-induced deformation across the excited volume[2]. In addition, electric fields and tip-induced strain gradients can cause changes in the local concentration of free ions in ion-conducting solids, thereby expanding or contracting the local volume and thus giving a piezoelectric-like deformation caused by voltage. This effect is at the base of so-called electrochemical strain microscopy[5,9,10].

Another relevant electromechanical coupling mechanism in PFM is flexoelectricity[11,12]. Direct flexoelectricity is a property allowed by symmetry in all materials, and it describes the appearance of polarization in response to a strain gradient. Converse and inverse flexoelectricity are the reverse phenomena of strain induced by polarization gradient[13] and strain gradient induced by polarization[14]. The inverse and converse flexoelectric effects have been experimentally demonstrated by, respectively, applying a voltage to a capacitor and measuring its bending[14,15] and by applying a voltage across a truncated pyramid so as to generate an inhomogeneous electric field inside it, thus causing the sample to deform[16–20].

Converse flexoelectricity, like direct flexoelectricity, is allowed by symmetry in all materials—including non-piezoelectric ones. Therefore, it should be a necessary ingredient of any PFM measurement because, when a voltage is applied to the tip, it generates an electric field that decays as we move away from its apex, resulting in an approximately radial electric field gradient. This electric field gradient must induce a converse-flexoelectric strain. Dividing the converse-flexoelectric strain by the voltage applied to the tip will yield a non-zero effective piezoelectric coefficient in any dielectric material. Here we demonstrate the existence of such converse flexoelectric effect in PFM, and show that it is quantitatively important, yielding significant piezoelectric-like response even if the material is ionically insulating and non-piezoelectric.

## Results

**Self-consistent simulation.** Flexoelectricity is a two-way coupling between polarization and strain gradient and, conversely, between strain and polarization gradient. Both direct and converse flexoelectricity are characterized by the same fourth rank tensor, and are hence allowed in centrosymmetric materials[12,21]. The converse-flexoelectric stress is expressed mathematically as:

$$\boldsymbol{\sigma} = \boldsymbol{\mu}\nabla\mathbf{E}, \tag{1}$$

where $\boldsymbol{\sigma}$ is the mechanical stress, $\mathbf{E}$ is the electric field and $\boldsymbol{\mu}$ is the fourth-order flexoelectric tensor. The excited volume below the PFM tip is subject to the converse flexoelectric effect due to the inhomogeneous nature of the electric field emanating from the tip (see Fig. 1). According to the Coulomb's law, the electric

field would decay as we move away from the tip, in a manner approximately equal to $r^{-2}$ (this field profile is modified by the contrast between the susceptibility in the atmosphere and inside the sample, but the physical principle still stands). The gradient of this electric field must induce a strain via the converse flexoelectric effect, see Eq. (1).

The exact solution of the problem is complicated due to complex interactions between the tip, the surface and the electromechanical response of the sample. In addition, the shape and size of the contact area is itself a complex function of the amount of force with which the tip is pressing on the surface, the exact tip shape and the elastic properties of both the tip and the sample. Moreover, the applied tip pressure induces strain gradients in the contact region. These also induce polarization by direct flexoelectricity. This polarization self-consistently modifies the electric field in the material by direct flexoelectricity and hence affect the nominal converse flexoelectric response.

In the absence of an analytical solution for this complex problem, we examine the role of converse flexoelectricity on the PFM response using a self-consistent computational model[22], based on a linear continuum theory of piezoelectricity[23], augmented with flexoelectricity. In this model, direct and converse flexoelectricity are intimately intertwined manifestations of the same coupling. The total electromechanical energy ($\mathcal{H}$), resulting fourth-order continuum equations and a mesh-free discretization method for solving numerically these equations are described in Supplementary Note 1.

In this work, the AFM tip is approximated as an ideal rigid sphere in contact with an ideal flat surface, see Fig. 2a. To model a frictionless contact, we follow the well-known Signorini-Hertz-Moreau model[24,25], which allows us to either control the tip indentation and measure force, or control force and measure the displacement of the indenter (cf. Supplementary Note 2). We consider the strong indentation limit for the electrical boundary conditions on the contact area[26], which implies that the electroelastic response of the sample dominates over electrostatic tip-surface interactions. The strong indentation limit assumption is valid for an applied force of $F > 100\,\text{nN}$, commonly used in PFM, for a typical tip radius of 50–100 nm[26]. In this limit, the electrical boundary conditions are:

$$\phi = V, \quad 0 \leq r \leq a \tag{2}$$

$$D_z = 0, \quad r > a \tag{3}$$

where $a$ is the contact radius, $\phi$ is the electric potential, $V$ is the tip voltage, and $D_z$ is the vertical electric displacement. The bottom side of the model is connected to the ground, i.e., the electric potential is fixed to zero, see Fig. 2b, c. We assume the charge-free condition $D_\text{n} = 0$ for all other faces of the computational domain, where $D_\text{n}$ is the normal electric displacement. Additional higher-order boundary conditions also arise from flexoelectricity[22,27] (Supplementary Note 1).

An important feature of the response is the size effect, because flexoelectricity is caused by a gradient: the sharper the contact is, the stronger the gradient will be, and thus the larger the flexoelectrically-induced deformation[12]. Moreover, the contact area between tip and surface grows as a function of the amount of force with which the tip is pressed onto the surface. Accordingly, we can expect that, upon increasing loads, the contact area increases, and thus the AFM tip-induced electric field gradient in the sample decreases. Gradient-induced apparent piezoelectricity will, therefore, be inversely proportional to contact force. This feature emerges as a useful tool to differentiate the converse flexoelectric electromechanical response from the regular piezoelectric response or the electrochemical strain, for which no dependence on the contact area should be observed[5,28].

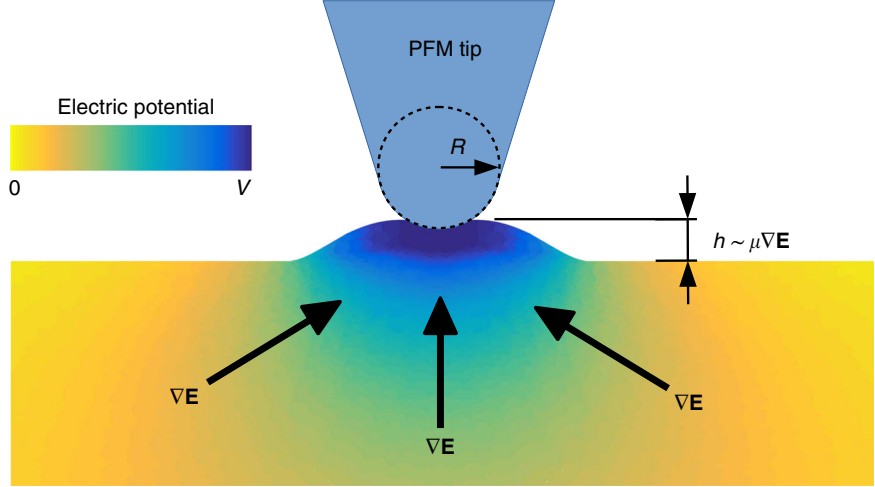

**Fig. 1** Schematic of piezoresponse force microscopy measurement. Piezoresponse force microscopy (PFM) operates by delivering a voltage $V$ to the surface of the material via an electrically conducting tip. In a piezoelectric material, the tip voltage $V$ will cause a local deformation, $h$, which is assumed to stem solely from the piezoelectric coupling. The effective piezoelectric coefficient is hence taken as $d_{33}^{eff} = h/V$. However, the tip voltage induces an inhomogeneous electric field below the PFM tip, which decays as we move away from the tip. The gradient of this electric field ($\nabla \mathbf{E}$) must induce a strain via the converse flexoelectric effect in all dielectrics, including non-piezoelectrics, see Eq. (1), which results in a measured deformation $h$, and consequently an apparent piezoelectric coefficient

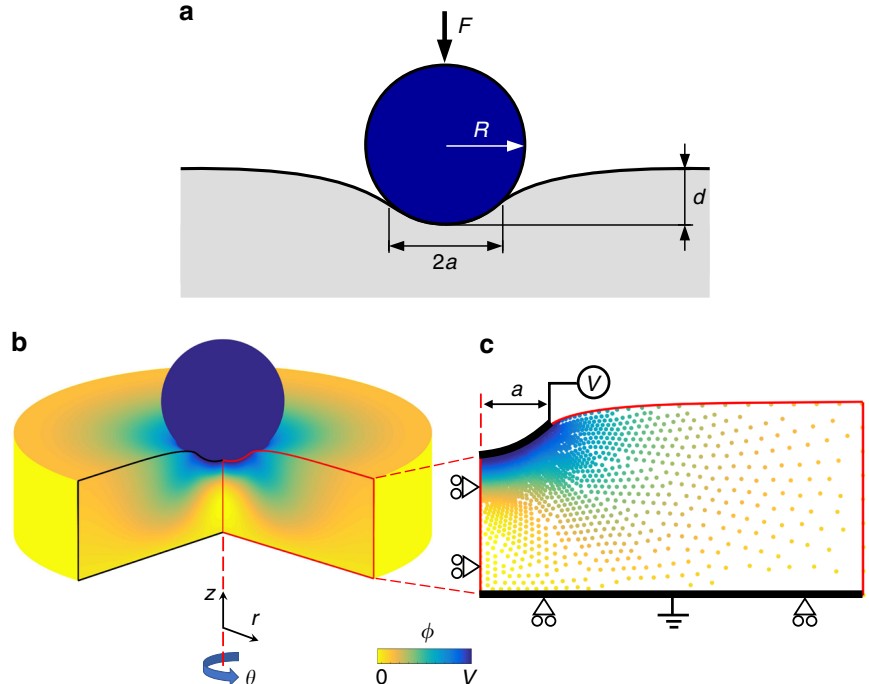

**Fig. 2** Contact of a spherical piezoelectric force microscopy tip with a flat surface. **a** Schematic of the contact of a spherical tip with radius $R$ under an applied load $F$ which induces an indentation depth of $d$ with contact radius $a$. **b** Axisymmetric model of the spherical contact. The rotational symmetry of the spherical tip-sample contact allows us to employ this two-dimensional axisymmetric model in a cylindrical coordinate system $(r, \theta, z)$. **c** Computational node set in the deformed configuration. The colour plot presents the distribution of the electric potential $\phi$. To capture the sharp changes of the strain and electric field in the excited volume, the computational nodes are distributed such that the nodal spacing is gradually diminishes as the contact surface is approached. The electric potential at the nodes in the contact surface ($0 \leq r \leq a$) is fixed to the tip voltage $V$ (voltage source symbol). The ground symbol indicates that the nodes on the bottom side of the model are connected to the ground, i.e., the electric potential is fixed to zero. The roller supports represent the mechanical boundary conditions which imply that the vertical and horizontal displacements are fixed on the bottom and left sides of the model, respectively

These qualitative predictions are quantitatively supported by numerical calculations. In Fig. 3, we show the simulation results for the effective piezoelectric coefficient as a function of contact radius and contact force, calculated for non-piezoelectric SrTiO₃.

The simulation protocol is as follows. We first progressively indent until reaching the desired force $F$ at $V = 0$, and then the force is kept constant as the tip is electrically biased and the induced displacement under the tip $\Delta u_z$ recorded. The effective

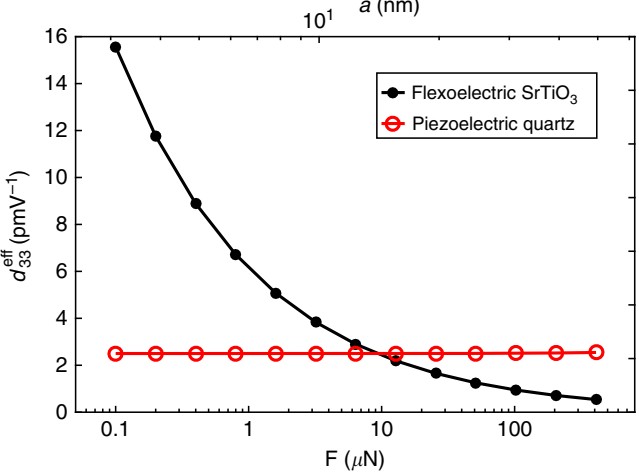

**Fig. 3** Effective piezoelectric coefficient as a function of contact radius $a$ and force $F$. The flexoelectric response is obtained for non-piezoelectric SrTiO$_3$ (STO) considering only flexoelectricity (both direct and converse). For comparison, we show the hypothetical response of an archetypal piezoelectric (quartz) assuming an absence of flexoelectricity. The contact radius is obtained under the applied force, in the absence of the tip voltage. The flexoelectric response shows a size-dependent behaviour which can be used as a means to qualitatively distinguish between flexoelectricity and piezoelectricity

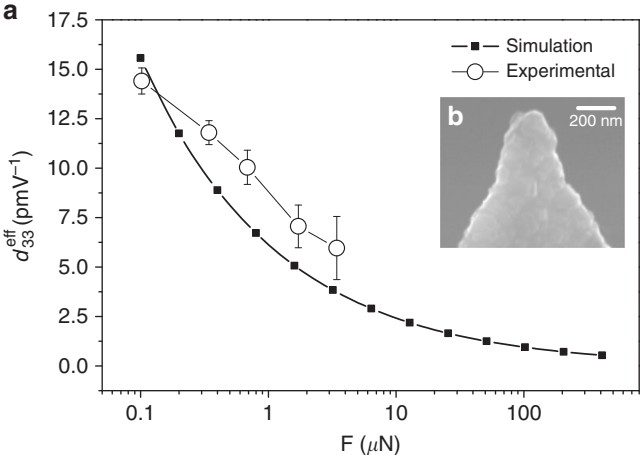

**Fig. 4** Study of converse flexoelectricity induced at the tip apex of an atomic force microscope cantilever as a function of the applied force. **a** Effective piezoelectric coefficient as a function of applied force for the SrTiO3 crystal. Filled squares correspond to the values obtained after the simulation. Empty circles correspond to the experimental values obtained with a Nanosensors CDT FM tip with a cantilever of medium stiffness ($k \approx 2.8 \, \mathrm{Nm}^{-1}$) coated with doped diamond. The error bars correspond to the error of the linear fitting of the experimental data, which correlates the measured electromechanical amplitude of oscillation $\Delta h$ with the $V_{ac}$ applied voltage. **b** The effective contact radius $a$ scales with the force, and is determined by the tip radius. The experimental tip radius is obtained after the measurement of the nanoscale electromechanical response from the scanning electron microscopy image of the used tip. In this case, the tip radius of the diamond coated tip is 105 nm, and is observed to keep a spherical shape after the measurements

piezoelectric coefficient is calculated as $d_{33}^{\mathrm{eff}} = \Delta u_z / V$, where $\Delta u_z$ is the tip vertical displacement under the tip voltage $V$, obtained from the simulation results (material parameters and details are given in Supplementary Note 2 and Supplementary Table 1). These calculations are compared against the simulation results obtained for a piezoelectric material quartz, neglecting flexoelectricity and assuming an effective piezoelectric coefficient $d_{33}^{\mathrm{eff}} = 2.5 \, \mathrm{pmV}^{-1}$. As expected, the pure piezoelectric response shows no dependence on the applied force $F$. Therefore, one can use either force or contact area dependence as a means to qualitatively distinguish between flexoelectricity and piezoelectricity. This flexoelectric size effect will be important in comparison with the experimental results.

**PFM experiment on non-piezoelectric dielectrics**. In order to validate our theoretical predictions, we have examined the piezoresponse of two different non-piezoelectric dielectrics, SrTiO$_3$ (STO) and TiO$_2$. The former is a cubic material with an effective flexoelectric coefficient of $\mu_{13}^{\mathrm{eff}} = 2.5 \, \mathrm{nCm}^{-1}$[29] while the latter has a rutile structure and an effective flexoelectric coefficient of $\mu_{13}^{\mathrm{eff}} = 1.7 \, \mathrm{nCm}^{-1}$[30]. The tips used were conductive diamond-coated Nanosensors CDT NCLR tips ($k \approx 72 \, \mathrm{Nm}^{-1}$, $R_{\mathrm{tip}} \approx 100 \, \mathrm{nm}$). In order to ensure that the applied pressure did not change the shape of the tip (and thus the validity of the contact model), we recorded high-resolution scanning electron microscopy images of the tips before and after the image. Using AFM, we also scanned the surface topography of the crystal before and after our piezoresponse characterization, to verify that there was no mechanical damage of the sample (Supplementary Note 5, Supplementary Figure 5). The effective piezoelectric coefficient $d_{33}^{\mathrm{eff}}$ as a function of tip load is shown in Fig. 4a for the STO crystal, and compared with the result of the theoretical calculations. The agreement is remarkable, considering that the calculation is done without any fitting of parameters. The qualitative trend of the curve as a function of force is also an important

evidence; if the piezoresponse of the STO sample were due to piezoelectricity[31], the piezoresponse should not depend on the indentation force (see Fig. 3). We note that Fig. 4b shows that the decrease in piezoresponse with increasing force is not due to a blunting of the tip, which remains spherical (with a relatively big radius due to its diamond coating) after the measurement. Meanwhile, the electromechanical response of TiO$_2$ is reduced by a factor of 0.74, as compared to STO, a proportion in agreement with the reduction of the corresponding flexoelectric coefficients (Supplementary Note 6, Supplementary Figure 6).

We also show that this response is not associated with electrochemical strain because the signal is not hysteretic and the temperature dependence is the opposite (decreases with temperature) of what one should expect if the origin was ionic conductivity (Supplementary Note 3–4, Supplementary Figure 2–4). We also discard a major contribution of the direct flexoelectric effect. The tip-induced strain gradient makes the material locally polar and thus piezoelectric, but this contribution has been observed to be proportional to the applied force: the bigger the tip pressure, the bigger the flexoelectrically-induced polarization[32,33]. In contrast, the experimental results in Fig. 4a show that the converse flexoelectric contribution is inversely proportional to the applied force, indicating that the dominant effect is the converse rather than direct flexoelectricity. The prevalence of converse flexoelectricity at low contact forces is also relevant because most PFM experiments are done under modest indentation forces of the order of 100 nN, which is the regime where our calculations and measurements show the strongest flexoelectrically-induced piezoelectricity.

## Discussion
The theoretical calculations, done without fitting parameters, show that apparent piezoelectric coefficients as high as 15 pmV$^{-1}$

can be measured in a cubic, non-piezoelectric material such as SrTiO$_3$. To put this number into context, this effective piezoelectric coefficient is about 6 times bigger than that of quartz, and similar to the piezoelectric coefficient of ZnO[34,35]. High-performance perovskite ferroelectrics, of course, have considerably larger piezoelectric coefficients (1–2 orders of magnitude bigger), but their piezoelectric coefficients are smaller in thin film form—which is the type of sample typically probed by PFM. Perovskite ferroelectric thin films typically display piezoelectric coefficients of the order of tens, exceptionally low hundreds, of pmV$^{-1}$, and thus converse flexoelectricity can represent a significant fraction of their total electromechanical response. Here it is important to emphasize that flexoelectricity can and does coexist with true piezoelectricity in ferroelectrics. This coexistence can lead to qualitatively distinct behaviour. In particular, the amplitude of the piezoresponse, which is independent of polarity when intrinsic piezoelectricity is the dominant effect, can be polarity dependent when flexoelectricity and piezoelectricity compete[36].

The situation, of course, is more dramatic when the material that is being studied is not truly piezoelectric, in which case the entire electromechanical signal arises from other physical phenomena. We emphasize, however, that the flexoelectrically-induced deformation is not an artefact: the effect is real, reproducible and inherent to any dielectric material. What is wrong is the interpretation of this voltage-induced deformation as piezoelectricity. The bottom line is that, since everything will look piezoelectric under a PFM, we cannot rely on this tool alone to determine whether a material is truly piezoelectric.

## Methods

**Self-consistent continuum model of flexoelectricity.** The self-consistent electromechanical field equations of flexoelectricity are a coupled system of fourth-order partial differential equations (PDEs) which demands at least $C^1$ continuous basis functions for a direct Galerkin method. To tackle the difficulty of solving these higher-order PDEs in a complex setup such as PFM, we resort to local maximum-entropy (LME) meshfree approximants[37]. The basis functions exhibit $C^1$ smoothness, and therefore a straight Galerkin approach is possible. The potential of the resulting computational model to simulate challenging setups, such as pyramid compression for quantifying flexoelectricity[20,22], fracture of ferroelectrics to reveal the fundamental manifestation of flexoelectricity in fracture physics[38], and piezoelectric bimorphs[39] unveiling complex interactions between piezo- and flexoelectricity, has been demonstrated. The details of this model are presented in Supplementary Note 1. We extend this model to contact problems to simulate the PFM experiment by considering the contact energy and its discrete form (Supplementary Note 2, Supplementary Figure 1).

**Experimental measurement of effective piezoelectric coefficient.** The effective electromechanical response was measured using a MFP 3D Asylum Research AFM. A $V_{ac}$ voltage was applied to the AFM tip at a frequency of 135 kHz and the obtained mechanical deformation $h$ of the surface was measured by the AFM cantilever net deflection out of resonance conditions. The deflection signal of the cantilever was externally analyzed with a SR844 Lock-In Amplifier to enhance the signal to noise ratio and the amplitude of the oscillation $h$ (pm) was recorded. To obtain the effective piezoelectric coefficient, the $V_{ac}$ voltage was applied following a triangular function with a $V_{ac}$ amplitude of ±10 V and a period of 80 s. The $d_{33}^{eff}$ was then calculated as the slope of the linear fit between the amplitude of the mechanical oscillation of a tip (deflection) as analyzed by the external lock-in amplifier and the excitation voltage, following the relationship $\Delta h = V_{ac} d_{33}^{eff}$.

## Code availability

The self-consistent computational model is implemented using an in-house C++ library. The library source code is available from the authors upon reasonable request.

## Data availability

All data presented in this work are available from the authors upon reasonable request.

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

## Acknowledgements

G.C. and N.D. acknowledge financial support from ERC Starting grant 308023, from Plan Nacional project MINECO Grant MAT2016-77100-C2-1-P and FIS2015-73932-JIN and support of Generalitat de Catalunya (Grant No. 2017-SGR-579). ICN2 acknowledges support from the Severo Ochoa Program (MINECO, Grant SEV-2017-0706) and the CERCA Programme / Generalitat de Catalunya. I.A. acknowledges the support of the Generalitat de Catalunya through the prize "ICREA Academia" for excellence in research and of the European Research Council (StG-679451). I.A. and A.A. also acknowledge the support of Generalitat de Catalunya (Grant No. 2017-SGR-1278).

## Author contributions

A.A. and N.D. contributed equally to this work. G.C. and N.D. performed and analysed the experiments. I.A. and A.A. developed the axisymmetric contact model and performed the simulations. G.C. wrote the paper with the help of all the other authors. All authors discussed the results, commented on the manuscript and gave their approval to the final version of the manuscript.

## Additional information

**Competing interests:** The authors declare no competing interests.

