## [Peer review file · Nature Communications]

Reviewers' comments:

Reviewer #1 (Remarks to the Author):

The manuscript intends to "demonstrate the existence of converse flexoelectric effects in PFM, and to show that they are quantitatively important, yielding significant electro-mechanical response in non-piezoelectric materials ". Although this may be relevant and important for PFM measurements, this reviewer did not see any significant breakthrough in fundamental understanding or materials development. This study presents theoretical calculations of the piezoelectric coefficient for SrTiO₃, which are standard calculation nowadays. Overall, I cannot recommend the publication of this manuscript based on the lack of significant advances mentioned above.

Reviewer #2 (Remarks to the Author):

This manuscript makes an excellent point, insofar as the electric field gradient, that is necessarily geometrically associated with the PFM measurement technique, must cause flexoelectric deformation, which could mistakenly be interpreted as piezoelectric distortion. The calculations and measurements on SrTiO₃ clearly show that this localised flexoelectric distortion can lead to a quite large inferred effective piezoelectric coefficient (especially under relatively low contact force conditions). Importantly, the modelling and the experiment show that the flexoelectric-induced apparent piezoelectric measurements have a characteristic fall-off with tip pressure (and associated tip-contact area) and so can be diagnosed and distinguished from true piezoelectricity.

I like the core message of this manuscript very much and think it adds significant insight for the many researchers using PFM to make semi-quantitative piezoelectric measurements on novel samples. I hence recommend it for publication. It may also give a neat method for the determination of flexoelectric coefficients at the nanoscale.

However, the authors should probably consider the following point before producing a final manuscript:

In the development of the model to allow the calculation of the flexoelectrically-induced distortion due to the electric field gradient, a great deal of emphasis is given to the manner in which the contact geometry changes with applied tip pressure. This is obviously important, as it changes the electric field distribution. However, increases in tip pressure will also induce strain gradient fields, which would cause the development of actual polarisation (again due to flexoelectricity), allowing the potential irony that a genuine piezoelectric distortion response could and should exist (from the electric field acting on the flexoelectrically-strain-gradient-induced polarisation). I looked for this additional level of complexity in the model, as I suspected the authors would expect it, given the emphasis on the mechanical contact and associated deformation, but I couldn't see it. Has this effect been included in the calculation of the effective d_{33} values ? If not, should it be ?

Reviewer #3 (Remarks to the Author):

The manuscript entitled "Converse flexoelectricity yields large piezoresponse force microscopy signals in non-piezoelectric materials" by Dr. Abdollahi and colleagues demonstrate converse flexoelectricity induced PFM signal. Since the appearance of polarization in response to a strain gradient, i.e. flexoelectricity, is well known, the converse flexoelectricity can be readily expected to contribute to the PFM signal. However, surprisingly, there is no such a report on the converse flexoelectricity induced PFM signal. Further, since there have been recently lots of discussions on the origin of the PFM signals, the present work can be very interesting for PFM as well as piezo/ferroelectric communities. Thus, this manuscript can be published in Nature Comm. However, the authors should consider following issues.

1. If tip load is changed, flexoelectric effect (not converse flexoelectricity) underneath the tip can be changed as well. How do the authors differentiate the converse flexoelectricity from the flexoelectricity? If it is not easy to differentiate, it may be better to take other ways to prove their claim. For example, piezoresponse as a function of electric field gradient.

2. In figure caption of Fig. 3, "the absence of flexoelectricity". Is it flexoelectricity or converse flexoelectricity?

3. In any cases of my second comment, there is no reason for excluding either flexoelectricity or converse flexoelectricity for quartz in Fig. 3 because, experimentally, both flexoelectricity and converse flexoelectricity may contribute to the PFM signal even for piezoelectric materials. I would strongly recommend to perform the similar experiment for the piezoelectric or ferroelectric materials such as quartz.

Reviewers' comments:

Reviewer#1 (Remarks to the Author):

The manuscript intends to “demonstrate the existence of converse flexoelectric effects in PFM, and to show that they are quantitatively important, yielding significant electro-mechanical response in non-piezoelectric materials”. Although this may be relevant and important for PFM measurements, this reviewer did not see any significant breakthrough in fundamental understanding or materials development. This study presents theoretical calculations of the piezoelectric coefficient for SrTiO₃, which are standard calculation nowadays. Overall, I cannot recommend the publication of this manuscript based on the lack of significant advances mentioned above.

We will not comment on the reviewer’s editorial recommendation. However, we do want to answer to this technical point: “*This study presents theoretical calculations of the piezoelectric coefficient for SrTiO₃, which are standard calculation nowadays*”. There are no calculations, standard or otherwise, of the piezoelectric coefficient of SrTiO₃ at room temperature. This is because SrTiO₃ is not a piezoelectric material at room temperature.

What our calculations show is that there is an APPARENT piezoelectric coefficient that appears when using piezoresponse force microscopy to characterize SrTiO₃ (or any other material, as illustrated by our measurement of the equally non-piezoelectric TiO₂). This piezoresponse is, as we show, unrelated to real piezoelectricity, being instead caused by converse flexoelectricity. We emphasize this point in the Abstract (Page 2) and in the last paragraph of the Discussion (Page 10) because it is the core message of our paper.

Reviewer #2 (Remarks to the Author):

This manuscript makes an excellent point, insofar as the electric field gradient, that is necessarily geometrically associated with the PFM measurement technique, must cause flexoelectric deformation, which could mistakenly be interpreted as piezoelectric distortion. (...) I like the core message of this manuscript very much and think it adds significant insight for the many researchers using PFM to make semi-quantitative piezoelectric measurements on novel samples. I hence recommend it for publication. It may also give a neat method for the determination of flexoelectric coefficients at the nanoscale.

However, the authors should probably consider the following point before producing a final manuscript:

In the development of the model to allow the calculation of the flexoelectrically-induced distortion due to the electric field gradient, a great deal of emphasis is given to the manner in which the contact geometry changes with applied tip pressure. This is obviously important, as it changes the electric field distribution. However, increases in tip pressure will also induce strain gradient fields, which would cause the development of actual polarisation (again due to flexoelectricity), allowing the potential irony that a genuine piezoelectric distortion response could and should exist (from the electric field acting on the flexoelectrically-strain-gradient-induced polarisation). I looked for this additional level of complexity in the model, as I suspected the authors would expect it, given the emphasis on the mechanical contact and

associated deformation, but I couldn't see it. Has this effect been included in the calculation of the effective d_{33} values ? If not, should it be ?

We thank the referee for this insightful comment. Indeed, the pressure-induced strain gradient does polarize the contact region due to direct (as opposed to converse) flexoelectricity. Once flexoelectrically polarized, the material would automatically behave as piezoelectric, since all polar materials are piezoelectric.

We note, however, that this effect is of higher order, and thus expected only at large fields and field gradients. Indeed, following the reasoning of the referee, the material would develop a strain-gradient-induced piezoelectricity. Mathematically, this would mean, focusing on polarization, that

$$\mathbf{p}_{\text{piezo}} = (\mathbf{e}_0 + \mathbf{e}_1 \nabla \boldsymbol{\varepsilon}) \boldsymbol{\varepsilon} \quad (\text{see Eq. S3 in Supplementary Note 1}),$$

where we have expanded the piezoelectric coefficient up to linear order in $\nabla \boldsymbol{\varepsilon}$. Here, we consider non-polar materials, and thus $\mathbf{e}_0 = 0$. The material parameter \mathbf{e}_1 expresses the strain-gradient-induced genuine piezoelectricity mentioned by the referee. Analogously, we would obtain for the mechanical stress an electric-field-gradient-induced contribution as:

$$\boldsymbol{\sigma}_{\text{piezo}} = -(\mathbf{e}_0 + \mathbf{e}_1 \nabla \mathbf{E}) \mathbf{E} \quad (\text{see Eq. S2 in Supplementary Note 1})$$

Note that this effect is distinct from flexoelectricity, which induces polarization following

$$\mathbf{p}_{\text{flexo}} = \boldsymbol{\mu} \nabla \boldsymbol{\varepsilon} \quad (\text{direct flexoelectricity, see Eq. S3 in Supplementary Note 1}),$$

and a mechanical stress following

$$\boldsymbol{\sigma}_{\text{flexo}} = \boldsymbol{\mu} \nabla \mathbf{E} \quad (\text{converse flexoelectricity, see Eq. S2 in Supplementary Note 1}).$$

Comparing these expressions, it is clear that the term involving \mathbf{e}_1 is quadratic in $\boldsymbol{\varepsilon}$ (or \mathbf{E}), and thus should in principle be a smaller effect than $\mathbf{p}_{\text{flexo}}$ (or $\boldsymbol{\sigma}_{\text{flexo}}$). While the mechanisms involving \mathbf{e}_1 in principle can exist, we note that its material coefficient \mathbf{e}_1 is distinct from the flexoelectric coefficients $\boldsymbol{\mu}$ and has not been experimentally characterized to the best of our knowledge. Our model, and all models of flexoelectricity that we are aware of, exclude such a higher-order effect.

Just for clarification purposes, we point out that our self-consistent treatment of flexoelectricity deals with direct and converse flexoelectricity in a unified way and on an equal footing [Abdollahi et al. *Journal of Applied Physics* **116**, 093502 (2014); Abdollahi et al. *Phys. Rev. B* **91**, 104103 (2015)]. This is apparent in the flexoelectric contribution in the energy (third term in Eq. S1 in Supplementary Note 1), which accounts for both direct and converse flexoelectricity. The direct flexoelectric effect appears in the constitutive equation for the electric displacement (the last term in Eq. S3 in Supplementary Note 1) and the converse flexoelectric effect manifests in the constitutive equation for the mechanical stress (the third term in Eq. S2 in Supplementary Note 1).

There is not a fundamental distinction between direct and converse flexoelectricity in our model, both being manifestations of the same physics. Our calculations account for the polarization induced by strain gradients caused by contact with the tip, which self-consistently modify the electric field in the material and hence affect the nominally converse flexoelectric response. Even if direct and converse effects are fundamentally intertwined, during PFM experiments an electric field gradient is introduced and

modified by the tip, and deformation of the material is recorded. Thus, it is meaningful to rationalize the phenomenology as converse flexoelectricity.

We have made this point explicit in the revised manuscript on Pages 4, 5 and 7. We have also added a comment on the experimental differentiation between the direct and converse flexoelectric contributions on Page 9, as mentioned also by another referee. In addition, to allow for an easier reading, we have moved the theoretical description of the contact model to the Supplementary Note 2.

Reviewer #3 (Remarks to the Author):

The manuscript entitled "Converse flexoelectricity yields large piezoresponse force microscopy signals in non-piezoelectric materials" by Dr. Abdollahi and colleagues demonstrate converse flexoelectricity induced PFM signal. Since the appearance of polarization in response to a strain gradient, i.e. flexoelectricity, is well known, the converse flexoelectricity can be readily expected to contribute to the PFM signal. However, surprisingly, there is no such a report on the converse flexoelectricity induced PFM signal. Further, since there have been recently lots of discussions on the origin of the PFM signals, the present work can be very interesting for PFM as well as piezo/ferroelectric communities. Thus, this manuscript can be published in Nature Comm.

We thank the referee for his/her reasoned endorsement.

However, the authors should consider following issues.

1. If tip load is changed, flexoelectric effect (not converse flexoelectricity) underneath the tip can be changed as well. How do the authors differentiate the converse flexoelectricity from the flexoelectricity? If it is not easy to differentiate, it may be better to take other ways to prove their claim. For example, piezoresponse as a function of electric field gradient.

Indeed, as the referee mentions, when the tip load is changed, so is the flexoelectrically induced polarization (direct flexoelectric effect). Likewise, the electric field gradients induced by the tip in turn induce stresses (converse flexoelectric effect). However, these phenomena are not independent: the flexoelectric polarization will interact with the applied electric field to determine the total electric field, and the converse flexoelectric stresses will interact with the applied loads to determine the stress and deformation state in the material. Our self-consistent model treats direct and converse flexoelectricity in a unified way and on equal footing. The 3rd term on the RHS of Equation S1 of the Supplementary Note 1 accounts both for direct and converse flexoelectricity.

Even though in the material these effects are intimately intertwined, it makes sense to speak about direct and converse effects when considering a specific measurement protocol. For instance, here, it is true that the tip-induced strain gradient polarizes the sample. This polarization, however, does not directly translate into a PFM signal, because tip pressure is static and field-induced deformation is dynamic –PFM measures only the ac response: oscillating deformation in response to the ac electric field.

The experimental differentiation between the direct and converse flexoelectric contributions is possible using different measurement protocol/mode of operation of the AFM. Direct flexoelectricity is proportional to applied force: the bigger the tip pressure, the bigger the flexoelectrically-induced polarization [Lu *et al.* Science 336, 59 (2012); Ocenasek *et al.*, Phys. Rev. B **92**, 035417]. In contrast, the converse flexoelectric contribution is inversely proportional to the applied force, because the electric field gradient becomes smaller as the contact area becomes bigger. The two force dependencies of the direct and converse effects are therefore opposite and allow for a direct identification of the dominant effect.

The experimental results in Figure 4 show that the induced piezoelectric coefficient is inversely proportional to force, indicating that the dominant effect is the converse flexoelectric contribution and not the direct one. We have added a note about this point in the revised manuscript on Page 9.

2. In figure caption of Fig. 3, "the absence of flexoelectricity". Is it flexoelectricity or converse flexoelectricity?

Both. Direct and converse flexoelectricity in our model are manifestations of the same physics. The caption has been changed to read: "For comparison, we show the hypothetical response of an archetypal piezoelectric (quartz) assuming an absence of flexoelectricity".

3. In any cases of my second comment, there is no reason for excluding either flexoelectricity or converse flexoelectricity for quartz in Fig. 3 because, experimentally, both flexoelectricity and converse flexoelectricity may contribute to the PFM signal even for piezoelectric materials. I would strongly recommend to perform the similar experiment for the piezoelectric or ferroelectric materials such as quartz.

This is an excellent point, and the referee is of course right: flexoelectricity can in theory coexist with piezoelectricity and may therefore also be present in quartz. For quartz, however, the flexoelectric coefficient is unknown but nanoindentation experiments have shown that, unlike in ferroelectrics such as BaTiO₃, it has no measurable effects [Gharbi *et al.*, International Journal of Solids and Structures 48 249–256 (2011)]. Our neglect of flexoelectricity in the piezoresponse of quartz, which was used only for illustration purposes in any case, is thus experimentally justified. Moreover, the purpose of Fig. 3 is to illustrate the size effect (dependence of the response on the contact area or force) of the flexoelectrically induced response which is not present in a purely piezoelectric response.

The point made by the referee nevertheless remains valid: the flexoelectric contribution to the total piezoresponse should in theory exist in piezoelectric materials. The coexistence of intrinsic piezoelectricity and flexoelectrically-induced piezoresponse, in fact, can yield qualitatively new behavior.

[redacted]

A paragraph has been added to the text along these lines in the revised manuscript on Page 10.

We thank all referees for their careful reading of our manuscript and the constructive feedback, which has fortified the paper. We hope that our revision has clarified the questions and remain at your disposal for any further clarification.

REVIEWERS' COMMENTS:

Reviewer #2 (Remarks to the Author):

The authors have taken my initial concern onboard and demonstrated that the true piezoelectric coupling between the stress-gradient-induced polarisation and the applied electric field is a second order effect and is, in any case, implicitly included in their treatment.

Responses to the other referees seem mature and constructive.

The article, as it stands is beautifully written and clear and I recommend publication.

Reviewer #3 (Remarks to the Author):

The authors have well addressed the issues raised by the reviewers and have reasonably revised the manuscript. Therefore, I would recommend its publication in Nature Communications.